# 3-Chloro-3-methyl-2,6-diarylpiperidin-4-ones as Anti-Cancer Agents: Synthesis, Biological Evaluation, Molecular Docking, and In Silico ADMET Prediction

**DOI:** 10.3390/biom12081093

**Published:** 2022-08-08

**Authors:** Arulraj Ramalingam, Nurulhuda Mustafa, Wee Joo Chng, Mouna Medimagh, Sivakumar Sambandam, Noureddine Issaoui

**Affiliations:** 1Department of Electrical and Computer Engineering, National University of Singapore, Singapore 117583, Singapore; 2Department of Medicine, Yong Loo Lin School of Medicine, National University of Singapore, Singapore 117599, Singapore; 3Cancer Science Institute of Singapore, National University of Singapore, Singapore 117599, Singapore; 4Department of Haematology-Oncology, National University Cancer Institute of Singapore, National University Health System, Singapore 119228, Singapore; 5Laboratory of Quantum and Statistical Physics (LR18ES18), Faculty of Sciences, University of Monastir, Monastir 5079, Tunisia; 6Research and Development Centre, Bharathiar University, Coimbatore 641046, India; 7BPJ College of Arts and Science, Kozhai, Srimushnam 608703, India

**Keywords:** piperidin-4-ones, myeloma, leukemia, lymphoma, molecular docking calculation, computational ADMET

## Abstract

Piperidine pharmacophore-containing compounds have demonstrated therapeutic efficacy against a range of diseases and are now being investigated in cancer. A series of 3-chloro-3-methyl-2,6-diarylpiperidin-4-ones, compounds (**I**–**V**) were designed and synthesized for their evaluation as a potential anti-cancer agent. Compounds **II** and **IV** reduced the growth of numerous hematological cancer cell lines while simultaneously increasing the mRNA expression of apoptosis-promoting genes, p53 and Bax. Molecular docking analyses confirmed that compounds can bind to 6FS1, 6FSO (myeloma), 6TJU (leukemia), 5N21, and 1OLL (NKTL). Computational ADMET research confirmed the essential physicochemical, pharmacokinetic, and drug-like characteristics of compounds (**I**–**V**). The results revealed that these compounds interact efficiently with active site residues and that compounds (**II**) and (**V**) can be further evaluated as potential therapeutic candidates.

## 1. Introduction

The piperidine ring is present in a variety of alkaloid natural compounds and is employed in drug applications because of its universal structural characteristics. According to Watson et al., numerous piperidone compounds have lately been employed in medicinal and preclinical research [1]. Piperidine-4-ones and its derivatives are a prominent class of biologically important N-heterocycles. Piperidine-containing compounds have a wide spectrum of biological characteristics, including anti-proliferative [2], anti-Alzheimer [3] and antiviral [4]. Furthermore, several saturated piperidine-2-ones have been used as critical chiral intermediates in the synthesis of naturally produced anti-cancer medicines [5]. Piperidine compounds have been synthesized using a variety of synthesis techniques due to their high biological characteristics [6,7]. Piperidine chemical compounds have been found in numerous studies to kill cancer cells in a variety of methods. It works by inhibiting the JAK/STAT protein kinase because it maintains anti-apoptotic, proliferative, and differentiated signals also are essential for the formation of cancer in the blood.

Aberrant JAK2 fusion proteins have been characterized in hematological cancers like acute lymphocytic leukemia (ALL) [8], acute myeloid leukemia [9] often mediating constitutive STAT activation. Similarly, the STAT3 signaling pathway is capable of activating and cancerous in nasal-type NK/T-cell lymphoma [10]. STAT can also be constitutively activated from an external stimulus in blood cancers. For example, in multiple myeloma, paracrine cytokine stimulation occurs as a result of myeloma cells interacting with bone marrow, resulting in increased production of IL-6, which activates JAK2 and STAT3, all of which are required for proliferation and survival. [11]. Curcuminoids with the piperidine ring have also been shown to target a variety of additional pro-survival mechanisms, including NF-κB activation [12], a well-known oncogenesis mediator [13]. While there has been significant improvement in the treatment of hematological cancers over the years, relapsed and refractory tumors remain tough to treat.

The present invention is focused on discovering the structural activity relationship of the 3-chloro-3-methyl-2,6-diarylpiperidin-4-one compounds (**I**–**V**). These compounds have demonstrated cytotoxic effects in myeloma, leukemia, and natural killer t-cell lymphoma cell lines. Furthermore, molecular docking calculations was performed using iGemdock software [14] and the Discovery Studio visualization [15] to predict the biological activity of synthesized compounds (**I**–**V**) and their interactions with the target proteins. The goal of this study was to use the molecular docking approach to further understand the interactions involved in the inhibition of the different proteins listed in the Protein Data Bank (PDB) [16]. Each synthetic compound’s pharmacokinetic and drug-like properties were also investigated.

## 2. Materials and Methods

### 2.1. Synthesis of 3-Chloro-3-methyl-2,6-diarylpiperidin-4-ones [(I)–(V)]

A series of 3-chloro-3-methyl-2,6-diarylpiperidine-4-ones (Compounds **I**–**V**) were synthesized. Benzaldehyde, p-chlorobenzaldehyde, p-fluorobenzaldehyde, p-methylbenzaldehyde, p-methoxybenzaldehyde, 3-chloro-2-butanone, and ammonium acetate were purchased with 99 percent purity from Sigma-Aldrich and used without further purification. A mixture of 0.1 mol ammonium acetate, 0.2 mol aldehyde, and 0.1 mol 3-chloro-2-butanone (20.4 mL) was heated slowly in ethanol. After letting it cool to room temperature, it turned into a colloidal suspension, which was then dissolved in 250 mL of diethyl ether. The dissolved solution was then treated with 100 mL of concentrated HCl (38 percent). The 3-chloro-3-methyl-2,6-diarylpiperidine-4-ones hydrochloride was filtered and washed with a mixture of 1:1 ethanol and diethyl ether. By adding 30 mL of 10% aqueous ammonia, the light yellowish-based solution was extracted from an alcoholic solution. The solution was diluted with 200 mL of distilled water. The portion of the crude sample was dissolved in absolute alcohol (100 mL), and warmed till the sample dissolved. Then, 2.0 g of animal charcoal was added to the alcoholic solution. Using Whatman filter paper, the dissolved solution was filtered to produce a clear solution. The filtered solution was allowed to cool and crystallize for 48 h. The produced white crystals are collected for further analysis. The respective yields of compounds (**I**) to (**V**) were 80%, 75%, 65%, 78%, and 60%.

Figure 1 represents the synthetic scheme for compounds (**I**) to (**V**). The synthesized compounds were analyzed using FT-IR, ^1^H NMR, ^13^C NMR, and mass spectra. In addition, the structure of all synthesized crystal compounds was confirmed by XRD single crystal measurement. All of the analyzed crystal data can be found at the Cambridge Crystallographic Data Centre [CCDC reference Numbers: Compound (**I**): 1524979; Compound (**II**): 1524977; Compound (**III**): 1508672; Compound (**IV**): 1524978; Compound (**V**): 1849763] [17,18,19].

### 2.2. Analytical Data

#### 2.2.1. 3-Chloro-3-methyl-2,6-diphenylpiperidin-4-one [Compound (**I**)]

**FT-IR** (cm^−1^) (KBr): 749.57 (νC–Cl) cm^−1^, 1602.76, 1495.15 (νC–C), 1713.51 (νC=O), 3063.43, 3007.40 (νC–H), and 3333.64 (νN–H); **^1^H NMR** (400 MHz, CDCl_3_): 1.38 (s, CH_3_ proton), 1.66 (s, NH proton), 2.50–2.45 [dd, H(5a) proton], 3.44–3.39 [t, H(5e) proton], 3.87 [s, H(2) proton], 4.00–3.97 [dd, H(6) proton], 7.41–7.16 (m, aromatic protons); **^13^C NMR** (CDCl_3_, 400 MHz): 22.25 (CH_3_), 45.60 (C-5), 61.49 (C-6), 69.88 (C-2), 72.02 (C-3), 129.52–126.89 aromatic carbons, 142.27, 137.32 133.93 aromatic carbons (*ipso*), 202.69 (C=O); **LRMS**: (ESI) *m/z* 332.1 [M+H]^+^; Melting point: 121.3 °C; Elemental analysis: C 72.03 (72.11), H 6.00 (6.05), N 4.52 (4.67).

#### 2.2.2. 3-Chloro-2,6-bis(4-chlorophenyl)-3-methylpiperidin-4-one [Compound (**II**)]

**FT-IR** (cm^−1^) (KBr): 799.88 (νC–Cl) cm^−1^, 1596.88, 1491.72 (νC–C), 1715.63 (νC=O), 3047.68, 3009.09 (νC–H), 3325.87 (νN–H); **^1^H NMR** (400 MHz, CDCl_3_): 1.43 (s, CH_3_ proton), 1.70 (s, NH proton), 2.54–2.51 [dd, H(5a) proton], 3.45–4.0 [dd, H(5e) proton], 3.93 [s, H(2) proton], 4.06–4.03 [dd, H(6) proton], 7.50–7.33 (m, aromatic protons); **^13^C NMR** (CDCl_3_, 400 MHz): 21.92 (CH_3_), 45.24 (C-5), 60.54 (C-6), 68.92 (C-2), 71.31 (C-3), 130.55–128.04 aromatic carbons, 140.41, 135.41, 134.67, 133.93 aromatic carbons (*ipso*), 201.73 (C=O); **LRMS**: (ESI) *m/z* 368.1 [M+H]^+^; Melting point: 139.8 °C; Elemental analysis: C 58.59 (58.64), H 4.29 (4.37), N 3.67 (3.80).

#### 2.2.3. 3-Chloro-2,6-bis(4-fluorophenyl)-3-methylpiperidin-4-one [Compound (**III**)]

**FT-IR** (cm^−1^) (KBr): 767.12 (υC–Cl) cm^−1^, 1609.56, 1508.41 (υC=C), 1716.35 (υC=O), 3075.55, 3008.86 (υC–H), 3325.34 (υN–H); **^1^H NMR** (400 MHz, CDCl_3_): 1.42 (CH_3_ proton, s), 2.07 (NH proton, s), 2.53–2.49 (dd, H(5a) proton), 3.45–3.40 (dd, H(5e) proton), 3.94 (H(2) proton, s), 4.06–4.03 (H(6) proton, dd), 7.54–7.03 (Aromatic protons, m); **^13^C NMR** (CDCl_3_, 400 MHz): 21.93 (methyl carbon), 45.25 (C-5 carbon), 60.52 (C-6 carbon), 68.85 (C-2 carbon), 71.58 (C-3 carbon), 137.87–114.85 aromatic carbons, 163.92, 163.44, 161.95, 161.48 aromatic carbons (*ipso*), 201.97 (C=O); **LRMS**: (ESI) *m/z* 368.1 [M+H]^+^; Melting point: 136.1 °C; Elemental analysis: **C** 64.28 (64.39), H 4.76 (4.80), N 4.05 (4.17).

#### 2.2.4. 3-Chloro-3-methyl-2,6-di-*p*-tolylpiperidin-4-one [Compound (**IV**)]

**FT-IR** (cm^−1^) (KBr): 738.68 (νC–Cl), 1615.57, 1513.79 (νC=C), 1715.40 (νC=O), 3095.35, 3007.79 (νC–H), 3332.57 (νN–H); **^1^H NMR** (400 MHz, CDCl_3_): 2.45 CH_3_ protons (attached to the phenyl ring, s), 1.43 (CH_3_ proton at C-3, s), 1.70 (NH proton, s), 2.54–2.51 (H(5a) proton, dd), 3.45–3.40 (H(5e) proton, dd), 3.93 (H(2) proton, s), 4.06–4.03 (H(6) proton, dd), 7.50–7.33 (aromatic protons, m); **^13^C NMR** (CDCl_3_, 400 MHz): 22.22 (CH_3_) (attached to the phenyl ring), 21.37 (CH_3_ at C-3), 45.58 (C-5), 61.18 (C-6), 69.62 (C-2), 72.16 (C-3), 129.69–126.76 aromatic carbon atoms, 139.32, 138.56, 138.01, 134.32 aromatic carbons (*ipso*), 203.07 (C=O); **LRMS**: (ESI) *m/z* 360.2 [M+H]^+^; Melting point: 98.2 °C; Elemental analysis: C 72.98 (73.27), H 6.62 (6.76), N 4.18 (4.27).

#### 2.2.5. 3-Chloro-2,6-bis(4-methoxyphenyl)-3-methylpiperidin-4-one [Compound (**V**)]

**FT-IR** (cm^−1^) (KBr): 792.84 (νC−Cl), 1609.94, 1510.77 (νC=C), 1718.26 (νC=O), 3048.35, 3007.73 (νC−H), 3332.92 (νN−H); **^1^H NMR** (400 MHz, CDCl_3_): 3.74 (methoxy proton, s), 1.36 (CH_3_ proton, s), 2.45 (H(5e) proton, d), 3.41 (H(5a) proton, t), 3.84 (H(2) proton, s), 3.95 (H(6) proton, d), 7.40–6.81 (aromatic protons, m); **^13^C NMR** (400 MHz, CDCl_3_): 22.01 (CH_3_), 60.69 (C-6), 45.44 (C-5), 202.89 (C-4), 72.09 (C-3), 69.07 (C-2), 55.28 (OCH_3_), 159.78 and 159.37 aromatic carbons (*ipso*), 130.35, 127.87, 114.09, 113.31 (aromatic carbons); **LRMS**: (ESI) *m/z* 368.1 [M+H]^+^; Melting point: 119.5 °C; Elemental analysis: C 67.65 (67.75), H 6.08 (6.16), N 3.79 (3.89).

### 2.3. Spectral Characterization

**FT-IR spectrum**: The FT-IR spectrum was collected using a PerkinElmer Frontier FT-IR spectrophotometer. The synthesized crystal compounds were powdered with potassium bromide, and the diffuse reflectance technique was used to capture the spectra; **^1^H NMR spectrum**: The proton NMR spectrum was captured using a Bruker 400 MHz NMR spectrometer adjusted at 400 MHz. To make the samples, dissolve 20 mg of the synthesized compound in 0.5 mL of CDCl_3_ containing 0.03 percent TMS; **^13^C NMR spectrum**: Proton decoupled ^13^C NMR spectra were captured using a Bruker 400 MHz spectrometer set to 100 MHz. ^13^C NMR Spectra were measured using solutions made by dissolving 20 mg of the substance in 0.5 mL of CDCl_3_ adding a few drops of TMS as an internal reference: **Mass spectrum**: A low-resolution mass spectrum was recorded to use an Agilent 6130B Quadrupole LC/MS in ESI mode with an Agilent 1260 Infinity LC system. Thermo Scientific Hypersil 150 × 2.1 mm 5-micron column was used to analyze the all samples.

### 2.4. Biological Study

#### 2.4.1. Cell Culture and Cell Treatment

Dimethyl Sulfoxide (DMSO) was used as solvent to dissolve all the 3-chloro-3-methyl-2,6-diarylpiperidin-4-ones, compounds (**I**–**V**). Usually the piperidin-4-one compounds have low solubility but the modification of compounds increases its rate of solubility with maximum 1% concentration of DMSO. All experiments were performed with equivalent DMSO control concentration. H929 (Multiple Myeloma Cell Line) MV-4-11 (Acute Myeloid Leukemia Cell Line) were grown in (RPMI)-1640 medium (HyClone Laboratories Inc., Logan, UT, USA) supplemented with 1% (*v*/*v*) Penicillin/Streptomycin (Biowest, Rue de la Caille, Nuaille, France), 1% (*v*/*v*) L-Glutamine (Sigma-Aldrich, St Louis, MO, USA) and 10% (*v*/*v*) fetal bovine serum (FBS; Biowest) at 37 °C in a humidified atmosphere with 5% CO_2_. SNK1 (NK Lymphoma Cell Line) was cultured in IL-2 (700 U/mL) Artemis medium-2 (NTSBio, Japan), 2% human serum (Capricorn Scientific, Germnany).

#### 2.4.2. Cell Viability Assay

In brief, cells were plated in tissue culture plates (Corning Inc., NY, USA) at 25,000 cells per well and then treated with the synthesized drug compounds and concentrations for 48 h. Cell viability was measured at the endpoint using the CellTiter-Glo (CTG) Luminescent Cell Viability Assay (Promega, Madison, WI, USA) according to manufacturers instructions. The cytotoxicity was calculated by comparing the cell viability of each sample to that of solvent-treated cells.

#### 2.4.3. Quantitative RT-PCR Analyses

The Qiagen RNeasy Mini Kit was used to recover total RNA (Qiagen, Hilden, Germany). iScript RT Supermix was used to create the cDNA (Bio-Rad Laboratories, Hercules, CA, USA). Applied Biosystems QuantStudio3 and SYBRgreen master mix (Waltham, MA, USA) were used to perform real-time quantitative PCR (qPCR). Calculations were based on the ΔΔCt method employing 18S transcript for normalization.

#### 2.4.4. Statistical Analyses

Data is presented as the mean value of *N* = 3 to 5 experiments with a ±standard deviation (SD). Statistically significant differences between treatment and control groups were assessed by one-way analysis of variance (ANOVA), followed by a Bonferroni post hoc test. *p*-value ≤ 0.01 was considered significant for all analyses.

### 2.5. Computational Study

#### 2.5.1. Molecular Docking

Molecular docking is computational technique to calculate the binding affinity and interaction energy of receptor-ligand complex. Docking study was performed of the synthesized compounds (**I**–**V**) by iGEMDOCK [14]. This method is a procedure that predicts the orientation of a molecule to another; it allows finding the orientation that maximizes the interaction even as minimizing the total energy score of the studied complex. In all the conditions of docking calculation, we used the settings as follows: population size is 800, the number 10 of generations is 80, and the number of solutions is 10. The RCSB protein data database was used to acquire the protein codes [16]. The docking findings were visualized using the discovery studio visualizer [15] package.

#### 2.5.2. ADMET Property Prediction

The SwissADME program [20] was used to determine the physicochemical properties and bioavailability of the synthesized compounds (**I**–**V**). The synthesized compounds (**I**–**V**) adhere with Muegge (Bayer) [21], Ghose [22], Egan (Pharmacia) [23], Veber (GSK) [24], and Lipinski’s rule of five [25].

## 3. Results and Discussion

### 3.1. Molecular Docking Study

The goal of molecular docking calculations is to estimate the most likely binding mechanism between such a protein and a ligand [26]. This technique is very useful in the drug design area since it narrows the choices of drugs for future in-vivo and in-vitro investigations [27]. In the current docking study, five compounds are docked with different kinds of proteins. All these compounds were tested in order to estimate their cytotoxic effect against the blood cancer cell lines: multiple myeloma, leukemia, NKT lymphoma. The examined compounds are docked with the proteins 6FS1, 6FSO (myeloma), 6TJU (leukemia), 5N21 and 1OLL (NK lymphoma). The crystal structure of the relevant proteins was picked up from the protein data bank (PDB) [16] and the docking simulations are carried out using iGemdock software [14]. All of the ligands (compounds **I**–**V**) and protein (PDB ID: 6FS1, 6FSO, 6TJU, 5N21 and 1OLL) in the best-docked position and their 2D interactions, which correspond to minimum energies, are represented visually. The results obtained from this docking calculation were analyzed to study the binding energy and the interactions of the docked structure (Table 1). It was observed that such energy is the total of the three interactions energies, which seem to be hydrogen bonds, electronic, and Van der Waals (VDW). From this table we notice that each protein with the five ligands has almost very similar energy values. Compound (**V**) with the proteins 6FSO, 6TJU, 5N21, and 1OLL are more stable than the other molecules because they had the best inhibition effect, with total energy scores of −95.79 kcal/mol, −87.11 kcal/mol, −109.10 kcal/mol, and −91.34 kcal/mol, respectively. Furthermore, the complex 6FS1-compound (**II**) has the highest binding energy (absolute value) −97.40 kcal/mol and the highest Van der Waals interactions −97.31 kcal/mol. We also notice from this table that the VDW interactions are the most dominant, more so than the hydrogen bonds and the electronic interactions. Furthermore, the two-dimensional visualizations depict the interaction between our compounds and the amino acids of the chosen proteins.

#### 3.1.1. Myeloma—Proteins: 6FS1 and 6FSO

Myeloma is the second most common cancer of the blood cancer that affects the hematopoietic bone marrow contained in the bones (Multiple myeloma is the second most common blood cancer characterized by an abnormal proliferation of malignant plasma cells in the bone marrow). Two proteins 6FS1 and 6FSO were utilized to design novel inhibitors against myeloma diseases. In the current study, we used the title compounds to assess their ability to suppress cancer cells. In this context, the total energies of 6FS1 in the following compounds listed in Table 1, were calculated using molecular docking data: compound (**I**), (**II**), (**III**), (**IV**) and (**V**) are −92.28, −97.40, −93.16, −89.41 and −89.33 kcal/mol, respectively (Figure 1 and Figure 2). As is clearly seen, the complex 6FS1—compound (**II**) possess the strongest binding affinity since it possesses the strongest energy (in absolute value) −97.40 kcal/mol, the strongest Van der Waals interaction −97.31 kcal/mol in comparison to the other complexes. Furthermore, the compound (**IV**) does have the lowest binding ligand −89.41 kcal/mol with the weakest VDW (−89.29 kcal/mol) interaction.

Likewise, the 6FSO-PCBE3C2B presents the best score of total energy (−92.75 kcal/mol) with the biggest VDW interaction (−92.65 kcal/mol). The two-dimensional representations (Figure 1) show the interactions among our compounds and the amino acids of the chosen proteins 6FS1 and 6FSO. Initial investigations showed that the protein 6FS1 with compounds (**I**–**V**) had only two hydrogen bonds interactions with some common amino acid residues such as 6LYA A: 271 and A: ARG: 263. Also the phenyl rings were bridged to Pi-sigma, Pi-alkyl and Pi-Pi T-shared but the VDW interactions doesn’t exist. The docked compounds on 6FSO possess six hydrogen bonds. Three hydrogen bonds are formed between the oxygen atoms of the compound BE3C2B (A: GLY: 104, A: TYR: 106 and A: GLY: 105), two typical hydrogen bonds were established between A: ARG: 380 and A: ARG: 369. Another bond was observed between the A: LEU: 54 amino acid and nitrogen atom of the compound PCBE3C2B. Furthermore, we observe VDW connections between the oxygen atoms of the ligands (**IV**) and (**V**).

#### 3.1.2. Leukemia—Protein: 6TJU

Leukemia is a cancer defined by the rapid and abnormal growth of blood cells which can be lethal if not aggressively treated. In recent years, the primary goal of cancer research has turned to the study of transmission, gene regulation, and genetic alterations in different cancers [28]. It is important to continue to discover new therapeutic targets and strategies as the disease can still relapse and become refractory after prolonged treatment. This study sought to investigate the potential therapeutic efficacy of novel compounds **I**–**V** for the treatment of leukemia cells. As a result, the five compounds were molecular docked using the 6TJU enzyme (shown in Figure 3). According to the docking calculations, the compound (**V**) has the best binding energies when compared to the other ligands. We have mentioned that this ligand has a large hydrogen bonds (−12.18 kcal/mol) with a powerful Van der Waals interactions (−74.93 kcal/mol). The binding site of the five proteins with the 6TJU ligand, followed by a two Dimensional diagram, reveals the presence of VDW interactions between the oxygen atoms of the ligands (**IV**) and (**V**). We note that the VDW (A: ARG: 58, A: SER: 44 and A: LYS: 57) and hydrogen bonds interactions (A: ARG: 58 and A: TRP: 53) are higher in the proteins 6TJU. The phenyl rings of the various compounds is implicated in A: GLU: 34 (Pi-anion), A: TYR: 13, A: TRP: 53, A: VAL: 43, A: SER: 46, A:TRP: 53 (Pi-Pi-T-shared), A: ALA: 48, A:ARG: 58, A: TYR: 13, A: ARG: 58, A:LYS: 57, A:LEU: 56 (Pi-Alkyl) and A: ARG: 54 (Pi-sigma) amino acid residues.

#### 3.1.3. NKT Lymphoma—Proteins: 1OLL and 5N21

Natural Killer T cell Lymphoma (NKTL) is a rare and aggressive form of Epstein Barr virus associated blood cancer of NK or T cell origin which is predominantly diagnosed in East Asian and South American populations. Here, in order to explore the potential efficacy of compounds **I**–**V**, the main interactions with target enzymes 1OLL and 5N21 were assessed with molecular docking simulations (Figure 4 and Figure 5). Table 1 summarizes the binding energy in ascending order of energy score of 5N21 on front of the five ligands was (**V**) > (**III**) > (**IV**) > (**I**) > (**II**) with interaction energy −109.10, −99.01, −97.50, −95.51 and −93.91 kcal/mol. As demonstrated, compound (**V**) has the highest score, with a good VDW interaction of −108.97 kcal/mol. The compound (**V**) has significant binding energy of approximately −91.34 kcal/mol for protein 1OLL with good VDW interaction of −91.21 kcal/mol. The 2D interactions plots of proteins 5N21 and 1OLL with ligands indicate the different types of interactions that exist among our compounds and amino acid. The amino acid sequences A: ARG: 13, A: THR: 12, A: GLU: 115, B: HIS: 14, B: ALA: 52, A: ASN: 21, and A: HIS: 14 were discovered to be involved in the creation of traditional hydrogen bonds in 5N21 protein. Then the phenyl ring of the title compounds bonded to A: CYS: 53, A: HIS: 116, A: MET: 114, A: SER: 93, A: LEU: 19, B: ALA: 15, A: ALA: 15 and B: HIS: 116 (pi-alkyl, pi-sigma, pi-pi T-shaped). The compounds interact with A: THR: 163, A: ARG: 136, A: SER: 137, and A: GLY: 135 amino acids involved in hydrogen bonding in 1OLL protein. While, A: ARG: 161, A: LYS: 183 and A: GLY: 135 interact with our ligands (**I**–**V**) forming VDW interaction. As compared to other bonds, van der Waals interaction is most broadly established in all molecules. The establishment of these interactions robustly predicts for the stable formation of these complexes. Molecular docking in silico confirms that these novel compounds can successfully bind with target ligands thereby potentially mediating impactful biological outcomes. Furthermore, the results suggested that the various ligands with chosen target proteins will have a high total energy score. So, we can conclude from these results that the novel 3-chloro-3-methyl-2,6-(diarylpiperidin)-4-ones compounds will be able to induce cytotoxic effects in myeloma, leukemia and NK lymphoma cell lines.

### 3.2. Biological Activity

Molecular docking analyses confirmed that compounds can bind to 6FS1, 6FSO (myeloma), 6TJU (leukemia), 5N21, and 1OLL (NKTL). Thus, we hypothesized that the compounds may be able to disrupt survival signaling in cancer cells. The anti-cancer efficacy of Compounds **I**–**V** were tested in a series of hematological cancer cell lines; multiple myeloma (MM), natural killer T-cell lymphoma (NKTL) and AML. Cancer cell lines were treated with increasing concentrations of compounds **I**–**V** and cell viability assessed at a 48 h endpoint. Suppression of cancer cell survival can be observed starting from 1 mM in H929 and MV-4-11 cancer cell lines and with complete inhibition in cell proliferation observed at 5 mM in all cell lines (Figure 6, Figure 7 and Figure 8). A one-way ANOVA revealed that there was a statistically significant decrease in cancer cell survival when H929 cells were treated by compound **I** (F(3,11) = 176.2, *p* < 0.0001), compound **II** (F(3,11) = 212.3, *p* < 0.0001), compound **III** (F(3,11) = 101.5, *p* < 0.0001), compound **IV** (F(3,11) = 88.4, *p* < 0.0001) and compound **V** (F(3,11) = 105.7, *p* < 0.0001) as compared to DMSO control. Bonferroni post hoc analysis showed significant differences between 1 mM and 5mM treatment versus the DMSO control in all compounds except compound IV.

This was similarly observed in the acute myeloid leukemia cell line. MV-4-11 appears to demonstrate greatest sensitivity to the compounds. The one way ANOVA performed identified that there was a significant induction of cancer cell death when cells were treated with by compound **I** (F(3,13) = 245.3, *p* < 0.0001), compound **II** (F(3,13) = 987.1, *p* < 0.0001), compound **III** (F(3,13) = 1030.4, *p* < 0.0001), compound **IV** (F(3,13) = 109.5, *p* < 0.0001) and compound **V** (F(3,13) = 290.1, *p* < 0.0001) as compared to DMSO control. Bonferroni post hoc analysis revealed significant differences between 1mM and 5mM treatment versus the DMSO control in all five compounds tested.

The natural killer T cell lymphoma cell line SNK-1 demonstrated the least sensitivity to compounds **I**–**V**. ANOVA confirmed a statistically significant reduction of cell viability induced by all five compounds, compound **I** (F(3,14) = 75.2, *p* < 0.0001), compound **II** (F(3,13) = 148.6, *p* < 0.0001), compound **III** (F(3,13) = 76.5, *p* < 0.0001), compound **IV** (F(3,13) = 123.6, *p* < 0.0001) and compound **V** (F(3,13) = 98.7, *p* < 0.0001) as compared to DMSO control. Bonferroni post hoc analysis identified significant differences only between 5mM treatment versus the DMSO control in all compounds suggesting that these compounds are less effective in NKTL. NKTL patients frequently express the P-glycoprotein/multi-drug resistance 1 which is a cell membrane pump that has been shown to mediate the efflux of drugs from cancer cells thus rendering it multi-drug resistant [29,30]. This may potentially account for the poorer efficacy of compounds **I**–**V** in NKTL.

Apoptosis dysregulation is a critical step in tumor development that ultimately affects therapy responsiveness [31]. Drug-induced apoptosis is frequently mediated by intrinsic pathway signaling resulting in the activation of Bax. Translocation and oligomerization of Bax at the mitochondria results in permeabilization of the mitochondrial membrane thereby resulting in the release of the apoptotic component cytochrome c, which promotes the activation of caspase 9 eventually triggering cancer cell death [32]. Many biological functions, including cell growth, DNA repair, and angiogenesis, are known to be regulated by p53. Activation of p53 as a result of DNA damage from drug treatment or abnormal oncogenic signaling pathways can increase the expression of a variety of apoptosis-related genes [33]. More importantly, p53 has been demonstrated to directly activate Bax, resulting in mitochondrial membrane permeabilization and apoptosis [34].

The mRNA expression levels of p53 and Bax was studied via qRT-PCR to evaluate if the synthesized compounds can modulate the transcriptional levels of these pro-apoptotic factors. Compounds **II** and **V** were selected for qRT-PCR because they showed better anti-cancer efficacy in the cell viability assays. A role for piperidones has been reported in targeting the proteasome pathway [35]. Targeting the proteasome pathway through proteasome inhibitors has been an effective strategy in MM and we were keen to first understand the specific mechanism of action of the piperidone compounds in the multiple myeloma model H929.

In H929, compound **II** induces a dose dependent increase in the mRNA expression of p53 which is accompanied by the upregulation of the mRNA transcripts of the pro-apoptotic protein bax (Figure 9). Compound **V** induces a similar upregulation of p53 and bax mRNA expression which was detected at 1 mM concentration (Figure 9). A one-way ANOVA confirmed that there was a statistically significant increase in the mRNA expression of p53 (F(4,19) = 76.1, *p* < 0.0001), and bax (F(4,19) = 64.5, *p* < 0.0001). Bonferroni post hoc analysis revealed significant differences between drug treatment versus the DMSO control as indicated in Figure 9. Compounds **II** and **V** can mediate the transcriptional upregulation of p53 and Bax mRNA expression thereby suggesting a potential role for p53 in stimulating Bax-induced apoptosis. It is important to note that the therapeutic potential of these compounds will need to be further evaluated against a reference anticancer drug such as etoposide. It will be also be important to extend the cytotoxic analysis of compounds **II** and **V** in normal healthy cells such as peripheral blood mononuclear cells (PBMCs) to establish if these compounds exhibit a good therapeutic window for further development and application in cancer.

### 3.3. In Silico ADMET Study

#### 3.3.1. Pharmacokinetic and Drug-like Properties

Sugiyama defines drug-like qualities as physicochemical traits (for example, solubility, stability, so on) and biological properties (ADME-Tox) that are compatible with high therapeutic efficacy [36]. The phrase “drug-like chemical”, according to Lipinski et al. [25] refers to compounds having appropriate ADME/Tox characteristics and the ability to function in Phase I clinical trials. According to Borchardt [37], medicinal chemists are responsible for improving not just the pharmacological action of therapeutic compounds. Since the Lipinski “rule of five” was published in 1997, researchers have been focused on the drug-like characteristic criteria of lead compounds [38]. The main physicochemical characteristics investigated in these studies are (1) MW (molecular weight), (2) lipophilicity, (3) HBD and HBA (number of donors and acceptors of hydrogen bonds), (4) Number of rings, ROT (rotatable bonds), PSA (polar surface area), and (5) acid/base properties [39,40,41,42]. The goal of drug-like property research is to aid in the drug design of compounds with potentially favorable ADME/Tox characteristics at the early phases of drug development, lowering the failure rate and, to some extent, decrease the cost of drug research and development.

Medicinal chemists are interested in the interaction between molecular structure and physicochemical characteristics as novel drug research that complements biological activity in drug development and discovery. Since van de Waterbeemd et al. first proposed the idea of “property-based design” in 2001, the concept has gained traction [43]. Its application in drug development and discovery has gotten lots of interest [44,45,46,47,48,49,50,51,52]. The chemical structure of a drug’s molecule has an impact on its physical properties such as Molecular Weight, lipophilicity, aqueous solubility (S), permeability, etc. Furthermore, the ADME/Tox characteristics (drug-like characteristics) of a drug compounds are influenced by its physical features, such as metabolic stability, capacity to cross the BBB (blood-brain barrier), toxicity and PK (pharmacokinetics). Finally, a drug molecule’s physicochemical characteristics and ADME/Tox properties influence its pharmacodynamic efficacy.

We aimed to investigate the chemical structural characteristics of compounds (**I**) to (**V**) in this context. SwissADME program was used to predict the physicochemical characteristics and bioavailability of component (**I**)–(**V**). All ligands (compounds **I**–**V**) adheres with Lipinski’s rule of five, Muegge (Bayer), Veber (GSK), Ghose and Egan (Pharmacia). Table 2 contains complete pharmacokinetics profiles for compounds (**I**)–(**V**), including physicochemical characteristics, druglikeness, lipophilicity, pharmacokinetics, medicinal chemistry, and water solubility. For compounds (**I**–**V**), the number of hydrogen bond donors (HBD) was 1, and the number of acceptors of hydrogen bonds (HBA) was 2, 2, 4, 2, and 4, respectively, whereas the number of bonds that can be rotated was 2, 2, 2, 2, and 4. The calculated measurements of compounds (**I**–**V**) fell within an acceptable range. The bioavailability radar for compounds (**I**–**V**) was depicted in Figure 10. The colored zone indicates the appropriate physicochemical region for oral bioavailability. The optimum ranges for bioavailability for each feature are Lipophilicity: LIPO-XLOGP3 from 0.7 to +5.0, Size: molecular weight from 150 to 500 g/mol. Polarity: POLAR-TPSA from 20 to 130 Å 2, Insolubility: INSOLU log S (ESOL) less than 6. INSATU-Fraction of carbon atoms as in sp3 hybridization must be more than 0.25 and Flexibility: FLEX—less than 9 rotatable bonds [20]. The resulting data shows that all the synthesized compounds (**I**–**V**) display a series of optimal properties associated with high bioavailability. The compounds (**I**–**V**) under investigation have drug-like properties and taken together with our findings that compounds **I**–**V** demonstrate some anti-cancer efficacy in vitro, the compounds might be a good candidates for further therapeutic development.

#### 3.3.2. Gastrointestinal Absorption and Brain Penetration prediction [BOILED-Egg]

Many drug development failures were caused by inadequate pharmacokinetics and bioavailability. Two pharmacokinetic characteristics that were estimated at various stages of the drug discovery process are gastrointestinal absorption and brain permeability. To achieve this aim, the Brain Or IntestinaL approximate permeability technique (BOILED-Egg) was proposed as an effective predictive model based on small molecule lipophilicity and polarity calculations [53]. The same two physicochemical characteristics are used to make predictions for both brain and intestine penetration. Egan et al. presented an elegant balance between these two sorts of models, who created a specific knowledge based on lipophilicity and polarity to distinguish between molecules that are well-absorbed and those that are poorly absorbed [23]. On the basis of a plot of two calculated descriptors: PSA versus ALOGP98 [54,55], the demarcation exists in an area with good characteristics for gastrointestinal absorption. The simplicity of the notion, ease of comprehension, and direct translation into the design of the molecular are all advantages of this approach. As opposed to rule-based models, it not only defines a limit but also provides a clear picture of how far a molecular structure is from the optimal physicochemical region for successful absorption. The BOILED-Egg model provides a fast, simple, readily repeatable, and statistically unparalleled robust approach for predicting the good gastrointestinal absorption with brain permeability of small compounds that might use in drug discovery and development.

On the tPSA vs. WLOGP plot (Figure 11), the ellipse that best classifies the compounds (**I**)–(**V**) in the HIA dataset was constructed to include that many well-absorbed and as fewer less absorbed compounds as feasible. The white section (yolk) seems to be the physicochemical region of compounds that have a high possibility of becoming absorbed by the gastrointestinal region, a process known as human intestinal absorption (HIA permeation). The physicochemical zone containing molecules having the great chance of penetrating the brain, known as the Blood-Brian Barrier, is depicted in yellow (BBB permeation). As illustrated in Figure 11, the compounds (**I**)–(**V**) with obvious oral bioavailability were overlaid onto the BOILED-Egg. Molecules that are relatively polar (PSA < 79 Å^2^) and reasonably lipophilic (log *P* between +0.4 and +6.0) have a good chance of reaching the CNS. The calculated polarity (PSA) findings for compounds (**I**) to (**V**) were 29.10 Å^2^, 29.10 Å^2^, 29.10 Å^2^, 29.10 Å^2^ and 47.56 Å^2^ respectively. Compounds (I) to (V) have computed lipophilic values of 2.70, 3.20, 2.90, 3.20, and 3.25, respectively. In conclusion, the calculated BOILED-Egg image shows that our compounds (**I**)–(**V**) have good pharmacokinetics and bioavailability characteristics.

## 4. Conclusions

Several methods, including molecular docking simulations, are being used to explore potential therapeutic candidates in cancer. For this reason, docking was utilized to evaluate the interaction between novel ligand-protein complexes. As a result, many ligand-protein pair interactions with high total energy scores were discovered, indicating that the compounds are promising candidates for additional preclinical research. Compound (**V**) was more persistent than the other compounds because it inhibits with selected proteins 6FSO, 6TJU, 5N21, and 1OLL. We subsequently tested the anti-cancer efficacy of the five novel 3-chloro-3-methyl-2,6-diarylpiperidin-4-ones (**I**–**V**) compounds and found that these compounds can inhibit the survival of multiple myeloma (H929), acute myeloid leukemia (MV411), and natural killer T-cell lymphoma (SNK1) blood cancer cell lines at selected concentrations. Of all the compounds, compounds (**II**) and (**V**) demonstrated consistent and stronger anti-cancer efficacy across all cell lines and stimulated the transcriptional upregulation of p53 and Bax which may potentially lead to the induction of apoptosis. It is interesting that this corroborates with findings from in silico docking simulations, which highlighted that compounds (**II**) and (**V**) display more stable interactions and are hence most likely to influence a cellular signaling response. Furthermore, in silico pharmacokinetics analyses suggest that the pharmacological compounds (**I**–**V**) have considerable inhibitory effects, indicating that these compounds should be investigated further for cancer therapy.

## Data Availability

Not applicable.

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
