# Peer review of "3-Chloro-3-methyl-2,6-diarylpiperidin-4-ones as Anti-Cancer Agents: Synthesis, Biological Evaluation, Molecular Docking, and In Silico ADMET Prediction"

_biomolecules, 2022, doi:10.3390/biom12081093_

Round 1

Reviewer 1 Report

The submission presents data obtained for a series of 3-chloro-3-18 methyl-2,6-diarylpiperidin-4-ones [(I) – (V)] compounds considered as a potential anti-cancer agent. Two compounds showed promising results- they reduced the growth of haematological cancer cell lines and increased the expression of pro-apoptotic genes, p53 and Bax. Additionally, the authors performed molecular docking calculations, to understand the structure-activity relationship.

Below, I have several suggestions for improvement. Without them, the manuscript cannot be published.

MINOR:

-          -The font size and spaces in the text (spaces) should be the same (line 74)

-         - Avoid starting sentences with a number or abbreviation (line 217 and so on). All these sentences should be changed.

-          Authors should explain more precisely the meaning of “n” (line 223 and so on)

-          -The title of the OX axis should be changed (I suggest writing units in the square bracket and deleting the divisor (Figure 1 and so on)

-          -Standardize the names of cell lines (text vs figures) and the name of compounds (Figure 4 and its legend is poorly readable).

-        -  The names of the compounds were specified earlier in the text, therefore the authors should use the nomenclature as in the previous figures

-         - Standardize the titles of Figures (especially Fig.9)

-          -Table 2 is poorly readable, especially “Water solubility- I suggest improving it)

-        -  The authors should explain the abbreviation “AML” in line 203.

MAJOR. The weakness of the manuscript is the statistical analysis and its lack. It is not clear how many data points were analyzed and compared, how they were normalized, and how they were expressed (as an SD or SEM?) Please show proof of normal distribution around the mean in each group when using ANOVA. Show individual data points in each plot.

Author Response

Much appreciation and thanks to all the reviewers for their time and valuable comments. With reference to the above manuscript, I would like to submit my Revised Manuscript for your kind consideration. We have addressed the comments and fulfilled the recommendations given by reviewers. The queries and answers are listed individually and it is uploaded on the "Reply to reviewers”.

Reviewer 2 Report

The paper presents the design, synthesis, anticancer activity, molecular docking calculation and ADMET property prediction of five novel 3-chloro-3-methyl-2,6-diarylpiperidin-4-ones.

 Comments:

The authors should compare the anticancer activity of the compounds with that of the reference anticancer drug.

It is worth assessing the cytotoxicity of compounds towards normal cells.

References should be adapted to the journal's requirements.

Text formatting should be carefully checked.

The language should be modified carefully.

Author Response

(The authors gave the same response as above.)

Reviewer 3 Report

In this study, the authors synthesized 3-chloro-3-methyl-2,6-diarylpiperidin-4-ones [(I) – (V)] compounds and evaluated their anticancer potency via cell-based assays, quantitative RT-PCR assays, and in silico studies. Strength of the study is that there is an integration of in  silico and in vitro analyses.

Below are my minor feedbacks for the authors’ consideration:

1.     Lines 22-23 –“Molecular docking analyses confirmed that compounds can bind to 6FS1, 6FSO (myeloma), 6TJU (leukemia), 5N21, and 1OLL (NKTL).” - This statement is unclear/confusing. Please revise. Instead of listing the crystals, please indicate identifies of each target protein that the compounds can bind to.

2.     Lines 216-217 – “The compounds were broadly potent starting from 1mM ...” – Statement seems inaccurate or vague. Please recheck. Based on Fig 2, for example, all five compounds seem ineffective at 1 mM. Also, the compounds also apparently exhibited different potency levels in different cell culture models.

3.     Strictly speaking, whether there are statistically significant differences among the values presented in Figures 1-3 is not clear. So, I strongly encourage the authors to perform statistical tests, i.e., one-way-ANOVA followed by a suitable multiple comparison test (post-hoc test, e.g., Tukey’s, Duncan’s ), on the cell assay results in the three figures.

4.     The cell assays also lacked a positive control/reference compound. The authors should consider running the cancer cell assays with a known anticancer drug for comparison.

5.     Lines 222-234 – Repetitive information (“The cell proliferation … where n=3.”) which has been mentioned in M&M and then repeated in the captions of a number of figures can be omitted.

6.     Lines 232-233 – “Compounds I-V are cytotoxic in the MV-4-11, … after 48 hours of treatment with 0, 0.1, 1.0, and 5.0 mM.” – Please revise this statement in the figure caption. It is incorrect. It misrepresents the result that, at 0.1 mM, the five compounds were NOT cytotoxic at all.

7.     Line 237– “Compounds I-V are cytotoxic in Natural Killer T-cell lymphoma cell line SNK1.” – Please revise this statement in the figure caption. It is inaccurate. Based on the figure, the compounds were apparently only effective at 5 mM, although the authors should also reconfirm this via statistical tests as suggested above. The five compounds seemed NOT cytotoxic at 0.1 and 1.0 mM.

8.     Line 241 – “Figure 3. Compounds I-V are cytotoxic in multiple myeloma cell line H929.” – Similar issue as brought up in the last comment. Please revise this statement in the figure caption appropriately.

9.     Regarding the qRT-PCR analysis, it will be helpful to readers if the authors could explain briefly why they tested on H929 cells, instead of MV-4-11 cells where the compounds expressed a more consistently cytotoxic effect.

10.  Also, why were only compounds II and V tested in qRT-PCR experiment? Can the authors please explain in the text why the other three compounds were not tested?

11.  Line 256 – “… are docked with the proteins (6FS1 and 6FSO), leukemia (6TJU), and NK lymphoma (5N21 and 1OLL)” – Please revise the statement. The meaning of the statement is confusing. 6TJU, for example is a crystal, not a disease/cancer. Also, please indicate the identities of the target/receptor proteins in the crystals. 6FS1, for example, is a crystal comprising Mcl-1 protein co-crystalized with a specific inhibitor AZD5991. Not all crystals consist of only target/receptor proteins. Some consist of both target proteins and crystalized ligands/inhibitors too.

12.  Did the authors perform the redocking step to validate the settings used in their docking procedures? They can consider including the evidence in the main text/supplementary file.

13.  How are the binding affinities/stabilities of the five compounds when compared with the co-crystalized ligands/inhibitors? Or when compared with other well-established anticancer drugs which can be used as a reference/for comparison?

14.  Did the authors use or not use PyRx in their molecular docking? Please recheck lines 58, 183 and 258.

·       Line 59 – “… molecular docking calculations was performed using iGemdock software

·       Line 183 - “The iGEMDOCK [14] and PyRx 0.8 algorithms [20] was used to calculate the molecular docking ...”

·       Line 258 – “… docking simulations are carried out using iGemdock software.”

15.  Please provide more details in M&M regarding the settings used in molecular docking simulation to allow others/readers to reproduce the observation/repeat the procedure.

16.  Section 3.3.1:

·       The section begins with lengthy text that looks like literature review rather than direct discussion of the results/data obtained.

·       Despite the length of Table 2 and the large amount of data presented, very little discussion/interpretation of their significance/relevance is provided in this section.

·       Lines 409-413 - “The optimum ranges for bioavailability for each feature are … 9 rotatable bonds.” This chunk of info also lacks support from cited references.

Author Response

(The authors gave the same response as above.)

Round 2

Reviewer 1 Report

Please , decrease the font size in table (there are still some sequences moved to new line)